# Promoting Sustainable Coal Mining: Investigating Multifractal Characteristics of Induced Charge Signals in Coal Damage and Failure Process

**Jinguo Lyu [1,2,\*]**, **Shixu Li [1]**, **Yishan Pan [3]** and **Zhi Tang [1]**

[1] School of Mechanics and Engineering, Liaoning Technical University, Fuxin 123000, China; lshixu1999@163.com (S.L.); tangzhi0127@163.com (Z.T.)
[2] College of Mining, Liaoning Technical University, Fuxin 123000, China
[3] School of Physics, Liaoning University, Shenyang 110036, China; panyish_cn@sina.com
[\*] Correspondence: glvjinguo2005@163.com; Tel.: +86-188-4182-1630

**Abstract:** Monitoring and preventing coal–rock dynamic disasters are essential for ensuring sustainable and safe mining. Induced charge monitoring, as a geophysical method, enables sustainable monitoring of coal–rock deformation and failure. The induced charge signal contains crucial information regarding damage evolution, making it imperative and important to explore its temporal characteristics for effective monitoring and early warnings of dynamic disasters in deep mining. This paper conducted induced charge monitoring tests at different loading rates, investigating the multifractal characteristics of induced charge signals during the early and late stages of loading. It proposed the maximum generalized dimension $D(q)_{max}$, multifractal spectrum width $\Delta\alpha$, and height difference $\Delta f$ as multifractal parameters for induced charge signals. Additionally, quantitative characterization of coal damage was performed, studying the variation patterns of signal multifractal characteristic parameters with coal damage evolution. This study revealed the induced charge signal of the coal body multifractal characteristics in the whole loading process. In the late loading stage, the double logarithmic curve demonstrated some nonlinearity compared to the previous period, indicating the higher non-uniformity of the induced charge time series. $D(q)_{max}$ and $\Delta\alpha$ in the late loading stage were higher than those in the early stage and increased with loading rates. As coal damage progressed, there were significant jumps of $D(q)_{max}$ in both the early and late stages of damage, with larger jumps indicating richer fracture events in the coal. The width $\Delta\alpha$ showed an overall trend of increase–decrease–increase with coal damage evolution, while the height difference $\Delta f$ fluctuated around zero in the early stage of damage development but increased significantly during severe damage and destruction. By studying the multifractal characteristics of induced charge signals, this study provides insights for the early identification of coal–rock dynamic disasters.

**Keywords:** coal–rock dynamic disaster; induced charge; multifractal; sustainable monitoring

## 1. Introduction

With the depletion of shallow coal resources, many mines in China have shifted to deep mining. Deep coal seams are subject to high geostress and various mining activities, increasing the frequency and severity of coal–rock dynamic disasters, which pose a significant threat to safe mining [1–3]. Due to the demands of underground operation, the monitoring and early warning methods for coal–rock dynamic disasters based on geophysical techniques have shown clear advantages and experienced rapid development. These methods include microseismic, ground sound, CT wave, electromagnetic radiation (EMR), surface potential, magnetic field, and infrared radiation methods [4–11]. Induced charge monitoring technology, as a geophysical method, tracks changes in the electric field around coal and rock to assess their damage levels, providing precursor information related to coal–rock fracture and dynamic disasters. Therefore, analyzing induced charge time-series

signals can establish a theoretical foundation for the monitoring and early warnings of coal–rock dynamic disasters.

At present, the research on the induced charge signals of coal and rock has achieved significant scientific findings. For example, Pan et al. [12] created the underground charge induction monitoring equipment and pioneered the fundamental induced charge monitoring theory for early warnings of ground pressure impact. Xiao et al. [13] and Zhao et al. [14] conducted composite studies on the induced charge alongside other physical parameters during the coal and rock fracture process. Yang et al. [15] proposed a stress–charge–temperature (SCT) coupling model to aid in predicting coal–rock dynamic disasters. Xiao et al. [16] introduced the charge criterion method for assessing the impact tendency of combined coal–rock. Zhu et al. [17] investigated the distribution characteristics of the induced charge signals generated during the deformation and failure of coal–gas complexes, thereby extending the applicability of induced charge monitoring technology. While the above studies have significantly advanced experimental research and underground monitoring, the time-domain analysis of signals still requires further study, especially multifractal characteristics.

Several studies have demonstrated the presence of fractal characteristics in surface potential, EMR, and acoustic emission (AE) signals. Wei et al. [18] and Yao et al. [19] discovered the multifractal variation characteristics in EMR signals during the coal–rock deformation process and utilized multifractal dimensions to predict dynamic disasters. Li et al. [20] analyzed the R/S law of the surface potential signals and confirmed the surface potential time series exhibit multifractal characteristics. Liu et al. [21] examined the multifractal parameters of AE signals from coal–rock masses with different strengths, revealing that multifractal parameter variations reflect the failure mechanism and energy fluctuations.

Experimental studies have demonstrated that charge separation is a premise for EMR signals during the damage and failure process of coal and rock, indicating that the induced charge, EMR, and surface potential signals are part of the same electrical phenomenon [22]. Furthermore, the evolution of coal damage often involves discontinuous events, such as the closure of internal microcracks and the propagation of new cracks, leading to non-steady time-series variations in induced charge signals. The self-similarity in the damage and destruction processes of coal micro-elements represents a hidden correlation within the signals [23]. These findings suggest that induced charge signals obtained from experiments or underground monitoring contain rich information. However, some of their time-series characteristics may not always be evident, particularly in practical situations. Therefore, conducting multifractal analysis on small-scale induced charge signals is essential for understanding their fluctuation characteristics and comprehending the distribution law of signals with varying intensities within the system. This approach can also serve as a reference for underground monitoring.

The failure of loaded coal is essentially an evolving process characterized by a gradual increase in internal damage and the induction of structural instability. In practical mining operations, excessive mining rates alter the deformation and failure rates of coal, aggravating its damage progression and affecting the charge intensity. Building upon this understanding, this study conducted the induced charge signal monitoring test on coal subjected to different loading rates sourced from a mine in Fuxin, China. Moreover, multifractal characterization parameters were employed to analyze the trend changes during the loading process. Combined with damage mechanics theory, a characterization relationship based on induced charge accumulation was proposed to depict the evolution degree of multifractal characteristic parameters of induced charge signals with coal damage and failure. The outcome of this study provides some key insights into the monitoring of coal–rock instability and failure.

## 2. Multifractal Theory

The multifractal is a method used to illustrate the extent of statistical self-similarity or heterogeneity, revealing a fractal structure with different fractal dimensions across various scales. This structure is typically depicted by the multifractal spectrum $f(\alpha)$-$\alpha$. As a

commonly used parameter for describing multifractals, the multifractal spectrum provides probability information regarding subsets with identical distributions of singular points. The singularity index $\alpha$ denotes the fractal dimension of a specific interval within the fractal body. Intervals sharing the same $\alpha$ form a fractal subset, indicating that the multifractal contains a combination of multiple subsets with varying fractal dimensions.

### 2.1. The Method of Multifractal Spectrum Calculation

The time series of induced charges for coal damage forms a one-dimensional distribution curve, and its multifractal spectrum can be calculated using the box dimension method [24,25]. Initially, the induced charge signal sequence is denoted as $x(i)$ and partitioned into $N$ one-dimensional boxes with scale $\varepsilon$.

Let $P_i(\varepsilon)$ represent the probability distribution function of the average amplitude of all signals in the $i$-th small box when its scale is $\varepsilon$, as shown in Equation (1).

$$P_i(\varepsilon) = \frac{S_i(\varepsilon)}{\sum_{i=1}^{N} S_i(\varepsilon)} \tag{1}$$

where $S_i(\varepsilon)$ is the sum of the signal amplitude in the $i$-th small box, and $\Sigma S_i(\varepsilon)$ is the sum of amplitudes of the entire signal sequence.

Define a partition function $\chi_q(\varepsilon)$ as shown in Equation (2).

$$\chi_q(\varepsilon) = \sum_{i=1}^{N} P_i(\varepsilon)^q \sim \varepsilon^{\tau(q)} \tag{2}$$

where $\tau(q)$ is the mass index and $q$ is the weight factor in the range of $-\infty < q < +\infty$. Different $q$ values represent the proportion of the probability distribution function $P_i(\varepsilon)$ with different sizes in the partition function $\chi_q(\varepsilon)$. When $q < 1$, smaller probability distribution functions $P_i$ (i.e., low-value charge signals) dominate the contribution to $\chi_q$. Conversely, when $q > 1$, larger probability distribution functions $P_i$ (i.e., high-value charge signals) dominate the contribution to $\chi_q$. In actual calculations, a larger range of $q$ is not necessarily better, as long as it does not have a significant impact on the calculation results.

When Equation (2) holds, it indicates a power–law relationship between the partition function and the partition scale. $\tau(q)$ can be calculated by the slope of the logarithmic curve $\ln \chi_q(\varepsilon)$-$\ln \varepsilon$, as shown in Equation (3).

$$\tau(q) = \lim_{\varepsilon \to 0} \frac{\ln \chi_q(\varepsilon)}{\ln \varepsilon} \tag{3}$$

The generalized dimension $D(q)$ can describe the multifractal features. Equation (4) shows the calculation of the $q$-th order generalized dimension $D(q)$.

$$D(q) = \begin{cases} \frac{1}{q-1} \lim\limits_{\varepsilon \to 0} \frac{\ln \sum_{i=1}^{N} p_i^q(\varepsilon)}{\ln \varepsilon} & (q \neq 1) \\ \lim\limits_{\varepsilon \to 0} \frac{\sum_{i=1}^{N} p_i^q(\varepsilon) \ln p_i(\varepsilon)}{\ln \varepsilon} & (q = 1) \end{cases} \tag{4}$$

Plotting the generalized dimension curve $D(q)$-$q$, a series of fractal dimensions $D(q)$ under different $q$ values can be obtained. The greater the deviation from 1, the higher the fluctuation of the signal and the stronger the multifractal characteristics. Equations (5) and (6) can be obtained from $\tau(q)$-$q$ by Legendre transformation.

$$\alpha = \frac{\mathrm{d}(\tau(q))}{\mathrm{d}q} = \frac{\mathrm{d}}{\mathrm{d}q} \left( \lim_{\varepsilon \to 0} \frac{\ln \chi_q(\varepsilon)}{\ln \varepsilon} \right) \tag{5}$$

$$f(\alpha) = \alpha q - \tau(q) \tag{6}$$

The $f(\alpha)$-$\alpha$ curve represents the multifractal spectrum of the induced charge time series, reflecting its uneven internal properties.

### 2.2. Multifractal Characterization Parameters

The fractal dimension effectively captures the instability degree of one-dimensional time-series signals. Studies indicate that in the critical instability state of a disordered system, temporal–spatial evolution exhibits implicit complexity, leading to multifractal characteristics. Consequently, the generalized dimension curve $D(q)$-$q$ serves as a quantitative indicator of multifractal characteristics [25].

Moreover, the characteristic parameters of multifractal spectrum $f(\alpha)$-$\alpha$ reflect internal differences within the signal. Here, $\alpha$ represents subsets of the induced charge signal, with $\alpha_{\min}$ representing high-value induced charge signal subsets and $\alpha_{\max}$ representing low-value induced charge signal subsets. Hence, the multifractal spectrum width $\Delta\alpha = \alpha_{\max}$-$\alpha_{\min}$ shows the amplitude difference of the induced charge signal. A larger $\Delta\alpha$ value indicates greater unevenness and intensity in the induced charge signal.

Furthermore, $f(\alpha)$ denotes the frequency at which a subset of induced charge signals with singularity index $\alpha$ appears during the loading process. The multifractal spectrum height difference $\Delta f = f(\alpha_{\max}) - f(\alpha_{\min})$ represents the proportion of the peak value in the induced charge signal. $\Delta f > 0$ indicates the dominance of low-value induced charge signals and vice versa.

## 3. Monitoring Test of Induced Charge Signals during Coal Damage and Failure Process

### 3.1. Sample Preparation and Test System

To investigate the multifractal characteristics of induced charges, monitoring tests were conducted during the coal damage and failure process under different loading rates of 0.06, 0.12, 0.60, and 1.20 mm/min. As depicted in Figure 1, the raw coal utilized in the test was sourced from a mining area in Fuxin, Liaoning Province, China. To ensure the validity of the statistical data and minimize test errors, coal samples were chosen from the same coal block, possessing a smooth surface and no visible cracks. Additionally, the bedding direction was oriented perpendicular to the loading direction. The raw coal was cut into rectangular specimens measuring $50 \times 50 \times 100$ mm$^3$, and their surfaces were polished to ensure that the unevenness of the surfaces at both ends was less than 0.02 mm. The physical parameters of the coal specimens are show in Table 1.

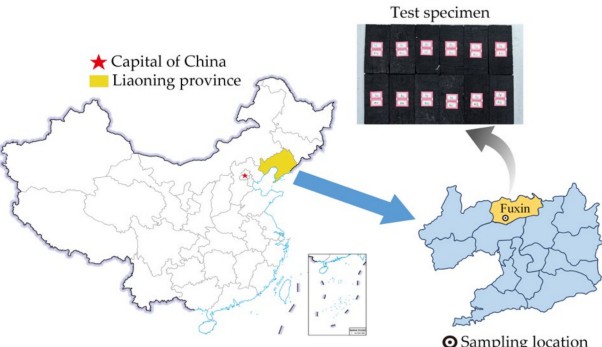

**Figure 1.** Specimen sampling location.

**Table 1.** Physical parameters of the test specimens.

| Density/(kg·m$^{-3}$) | Compressive Strength/MPa | Elastic Modulus/GPa |
| --- | --- | --- |
| 1176.28 | 6.36 | 0.66 |

The test system primarily consists of a stress loading system, data acquisition system, and electromagnetic shielding system, as shown in Figure 2. The stress loading system is a TAW-2000 electro-hydraulic servo pressure testing machine, capable of exerting a maximum axial load of 2000 kN. The data acquisition system comprises two non-contact induced charge sensors, a stress sensor, and a data acquisition terminal connected to a dynamic signal analyzer. The induced charge sensors are positioned on both sides of the

sample, 2.5 cm away, and the sampling frequency of the signal analyzer is set to 1 kHz. The electromagnetic shielding system employs dense copper mesh to shield the direct current (DC) power supply and dynamic signal analyzer, minimizing external noise interference. Before the test, the data acquisition system is turned on, and the uniaxial compression test is started when the environmental signal is stable. Since the stress sensor and induced charge sensors are connected to a dynamic signal analyzer, the signal changes during the test can be displayed in real time from the terminal.

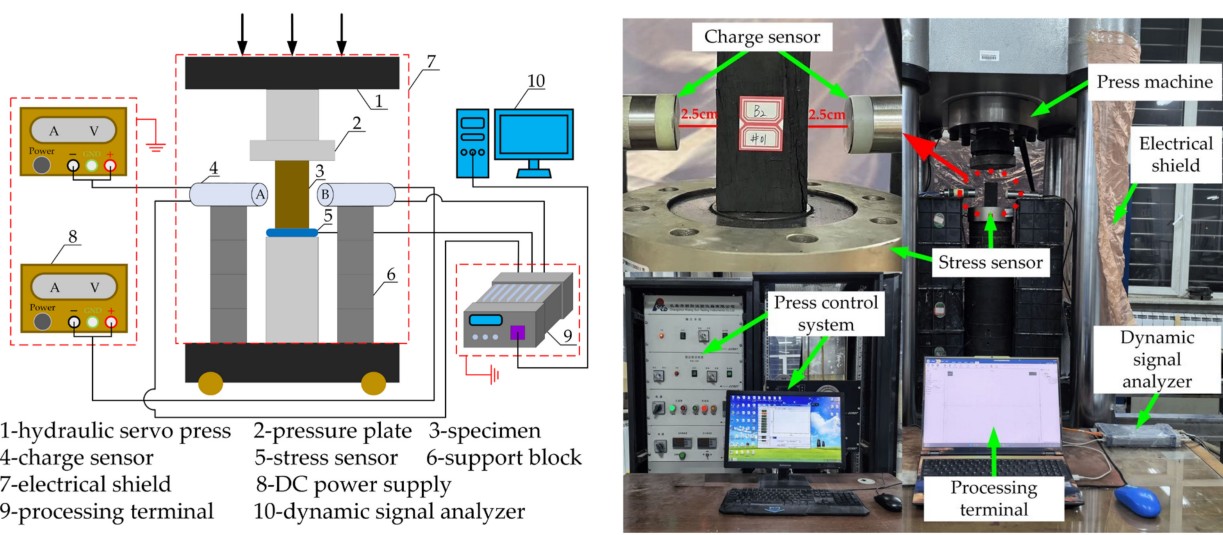

1-hydraulic servo press   2-pressure plate   3-specimen
4-charge sensor   5-stress sensor   6-support block
7-electrical shield   8-DC power supply
9-processing terminal   10-dynamic signal analyzer

(**a**) Test schematic        (**b**) Test equipment

**Figure 2.** Induced charge monitoring test of stressed coals.

### 3.2. Test Scheme

In this test, the displacement loading method was utilized, with the samples categorized into four groups for testing at various loading rates. Each group was subjected to testing on three coal samples, resulting in a total of 12 coal samples along with their corresponding induced charge signals and mechanical properties. The testing scheme for each group is detailed in Table 2.

**Table 2.** Test scheme.

| Specimen | Loading Rates/(mm·min$^{-1}$) |
| --- | --- |
| Group 1 | 0.06 |
| Group 2 | 0.12 |
| Group 3 | 0.60 |
| Group 4 | 1.20 |

### 4. Stress-Induced Charge Responses of Damaged Coal

The mechanism of induced charge generation reveals its association with piezoelectric and triboelectric effects, crack propagation, and the collapse of charged coal particles [26]. These effects weaken the binding effect of charges inside the coal, resulting in the release of free charge and the modification of the surrounding electric field. Representative test results at each of the four loading rates, denoted as A1, A2, A3, and A4, were selected for analysis. Figure 3 shows the relationship between stress and the induced charge over time. Generally, the induced charge signal tends to gradually increase as the loading progresses, displaying good synchronization with the stress. Fracture events occurring within the coal are often accompanied by a sudden drop in stress, followed by an immediate increase in the induced charge signal. Moreover, when the stress reaches the uniaxial compressive strength, the induced charge signal reaches its maximum value.

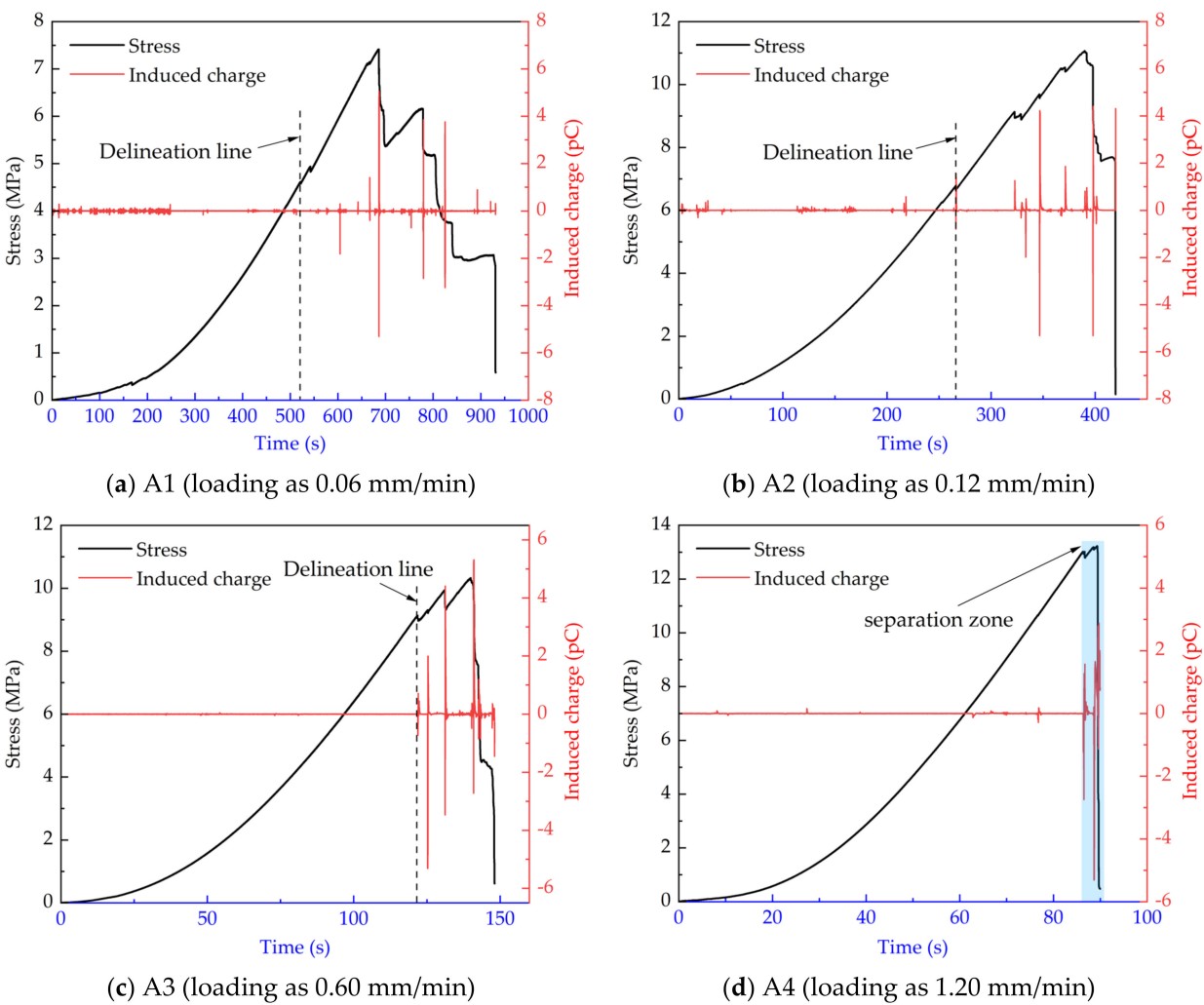

**Figure 3.** Time–stress–induced charge curves of coal samples at different loading speeds.

The uniaxial compression process of coal typically includes four stages: compaction, elasticity, plasticity, and failure. By using the starting point of the plastic stage of coal as the delineation line to distinguish between loading stages, namely the early and late stages (with only a separation zone marked for the A4 specimen due to its short plastic and failure stages), notable differences between these stages are apparent. During the early loading stages, primary microcracks gradually compress and close, resulting in the detection of only a small amount of low-value induced charge signals. Conversely, in the late stage of loading, primary and new cracks in the coal accelerate and converge to form a sliding surface, ultimately connecting to a fracture surface. The main failure occurs rapidly, accompanied by a relatively high-value induced charge signal characterized with a high degree of fluctuation. In addition, Figure 3 shows that if the loading rate is relatively low, primary microcracks in the coal are more fully developed during the early loading stage, and the low-value induced charge signal can be monitored multiple times. On the contrary, when the loading rate increases, the development time of fractures shortens, resulting in minimal free charge within the coal and no obvious signal detection. However, in the late stage of loading, with the continuous accumulation and evolution of coal damage, a higher amplitude of induced charge signal can be monitored.

## 5. Multifractal Characteristics of Induced Charge Signals

### 5.1. Scale Invariance of Signals

Each representative induced charge signal is divided into early and late loading time series, and the sequence is segmented into subsets of scale $\varepsilon$. The probability of each

subset is calculated to obtain the probability distribution $P_i(\varepsilon)$ of the induced charge signal sequence, and its partition function $\chi_q(\varepsilon)$ is solved. Figure 4 shows a double logarithmic curve between the partition function and the partition scale, with the weight factor $q$ ranging from −20 to 20, in increments of 1.

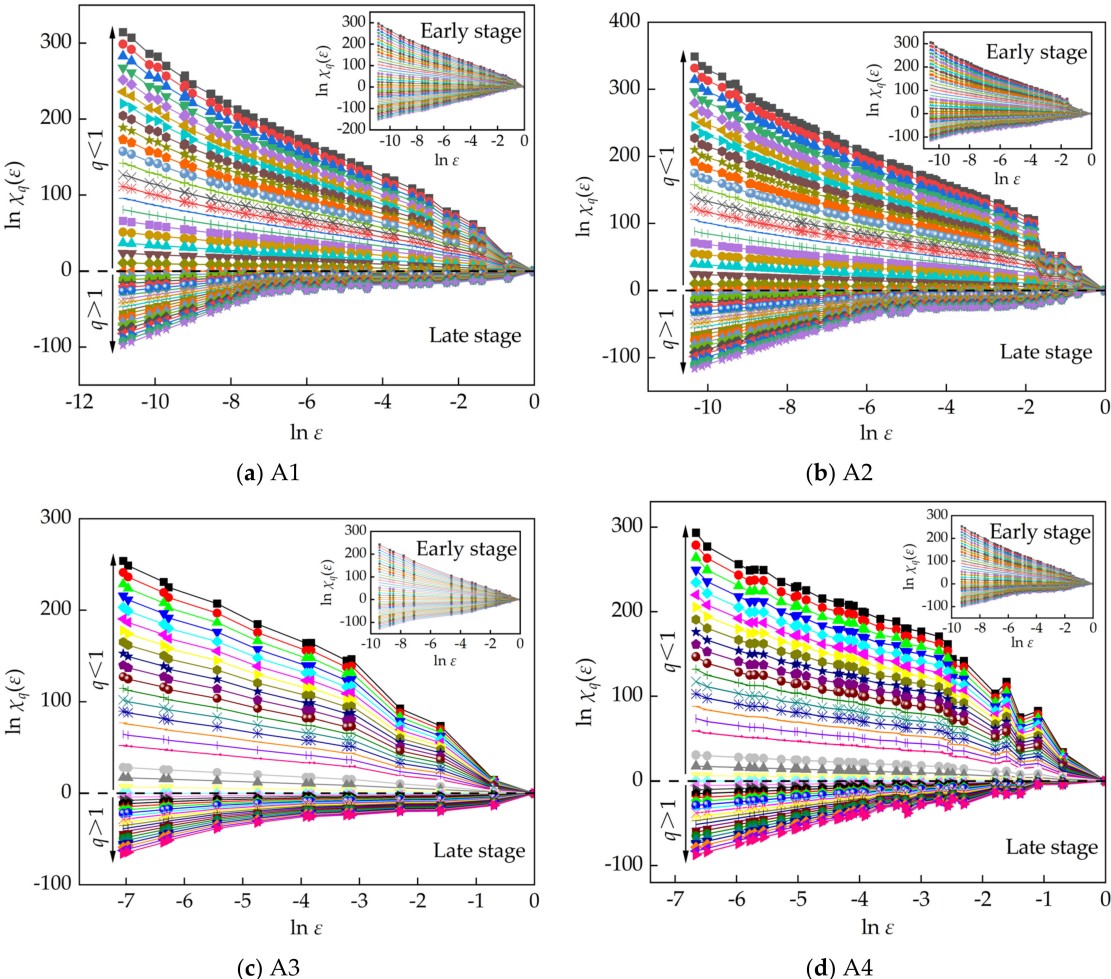

**Figure 4.** Double logarithmic curves of induced charge signal sequence at early and late loading stages.

The calculation results indicate that in the early stage of loading, the double logarithmic curve often shows a linear relationship, suggesting that the partition function $\chi_q(\varepsilon)$ and scale $\varepsilon$ satisfy a power–law relationship, indicating the scale invariance in the induced charge signals. This observation reflects that the waveform amplitude of the induced charge signal from large-scale coal is exponentially higher than that of small-scale coal during the early loading stage. However, in the late stage, the double logarithmic curve becomes progressively rugged, and its nonlinear characteristics are enhanced as loading rates increase. Additionally, with $q = 1$ serving as the dividing line, the lower curve is denser than the upper curve, implying that high-value induced charge signals hold a distinct advantage in the late stage, with the fluctuation degree intensifying with increases in the loading rate [27].

### 5.2. Multifractal Characterization Parameters of Signals

According to the multifractal methods, representative specimens were selected to calculate their generalized dimension $D(q)$ and multifractal spectrum $f(\alpha)$-$\alpha$ for the induced charge time series during early and late loading stages. Figure 5 displays the strictly decreased relationship with the increase in $q$, indicating the multifractal characteristics of the induced charge signal waveforms. The maximum generalized dimension $D(q)_{\text{max}}$, corresponding to the minimum weight factor $q$, is selected as the multifractal characteristic

parameter of the induced charge signals. This parameter reflects the maximum non-uniformity and the most significant multifractal characteristics of the induced charge signal waveform.

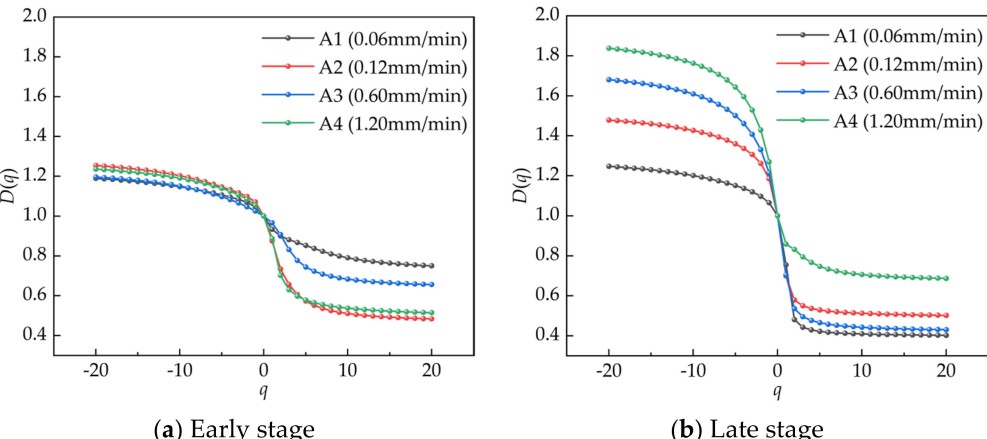

(**a**) Early stage　　　　　　　　　　　　　(**b**) Late stage

**Figure 5.** Generalized dimension $D(q)$-$q$ of induced charge signals at different loading stages.

The figure indicates that under the same loading rate, the generalized dimension $D(q)$ of the induced charge signal in the late stage is larger than that in the early stage, suggesting that the multifractal characteristics of the induced charge signal waveform resulting from the instability and failure of the coal body are stronger than the conventional induced charge signal waveform. Moreover, with the increase in loading rate, the $D(q)_{max}$ in the early stage of loading remains near 1.2, while the $D(q)_{max}$ in the later stage has a significant upward trend. When the loading rate reaches its maximum of 1.20 mm/min, the $D(q)_{max}$ also reaches its peak at 1.84. Therefore, by calculating the generalized dimension $D(q)_{max}$ of the induced charge signal time series, it can be effectively determined whether the coal has undergone a high-energy rupture event.

The multifractal spectrum $f(\alpha)$-$\alpha$ of the induced charge signal waveform is depicted in Figure 6, with its spectral parameters detailed in Table 3. The figure reveals that a wider $f(\alpha)$-$\alpha$ curve corresponds to a larger $\Delta\alpha$, showing a greater amplitude difference in the induced charge signal waveform.

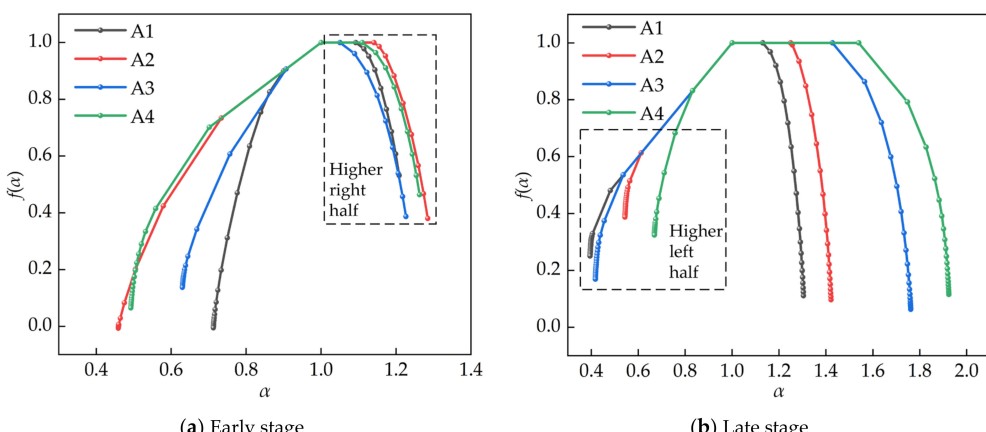

(**a**) Early stage　　　　　　　　　　　　　(**b**) Late stage

**Figure 6.** Multifractal spectrum $f(\alpha)$-$\alpha$ of induced charge signals at different loading stages.

In the early stage, the right half of the spectrum is higher, and $\Delta f > 0$, suggesting that the amplitude of the induced charge signal at this time is very low, primarily due to small fracture events inside the coal. However, in the late stage, the spectrum shows that the left half is larger, and $\Delta f < 0$, indicating the dominance of high-value induced charge signals. This occurrence is attributed to the accelerated damage and fracture of coal, resulting in the continuous generation of high-value charges.

**Table 3.** Multifractal spectrum parameters.

| Specimen | $\alpha_{\min}$ | $f(\alpha_{\min})$ | $\alpha_{\max}$ | $f(\alpha_{\max})$ | $\Delta\alpha$ | $\Delta f$ |
|---|---|---|---|---|---|---|
| A1-early | 0.7125 | 0.0005 | 1.2084 | 0.5321 | 0.4959 | 0.5316 |
| A2-early | 0.4598 | 0.0070 | 1.2845 | 0.3800 | 0.8247 | 0.3730 |
| A3-early | 0.6299 | 0.1370 | 1.2265 | 0.3871 | 0.5966 | 0.2501 |
| A4-early | 0.4923 | 0.0646 | 1.2627 | 0.4638 | 0.7704 | 0.3992 |
| A1-late | 0.3950 | 0.2509 | 1.3044 | 0.1111 | 0.9094 | −0.1398 |
| A2-late | 0.5434 | 0.3876 | 1.4217 | 0.0971 | 0.8783 | −0.2905 |
| A3-late | 0.4174 | 0.1696 | 1.7615 | 0.0634 | 1.3441 | −0.1062 |
| A4-late | 0.6689 | 0.3245 | 1.9241 | 0.1156 | 1.2552 | −0.2089 |

## 6. Multifractal Characteristic Variation in Induced Charge Signal during Coal Damage Process

While the maximum generalized dimension $D(q)_{\max}$ and the multifractal spectrum parameters $\Delta\alpha$ and $\Delta f$ jointly convey the singularity of the induced charge signal, their correlation with coal damage remains incompletely investigated. The process of damage and failure in loaded coal represent a nonlinear process in the time domain. Therefore, exploring the multifractal characteristics of statistical signals with varying degrees of coal damage could provide deeper insights into the relationship between these phenomena.

### 6.1. Characterization Relationship of Coal Damage Based on Induced Charge

According to the characteristics of the induced charge signal, combined with the statistical relationship of coal damage, the characterization relationship of coal damage based on induced charges is established.

The damage factor D is defined as the proportion of the failure micro-elements to the total micro-elements, which can be expressed by Equation (7) [28].

$$D = \frac{A_d}{A} \tag{7}$$

where $A_d$ refers to the damaged area of material and $A$ represents the initial area of material.

If the charge accumulation of the entire section $A$ of the coal mass is $Q_m$, the charge occurrence rate $i_q$ per unit area can be expressed by Equation (8).

$$i_q = \frac{Q_m}{A} \tag{8}$$

When the damaged area reaches $A_d$, the charge accumulation $Q_d$ can be calculated by Equation (9).

$$Q_d = i_q A_d = Q_m \frac{A_d}{A} \tag{9}$$

Substituting Equation (9) into Equation (7), we can obtain Equation (10).

$$D = \frac{Q_d}{Q_m} \tag{10}$$

Therefore, the charge accumulation can characterize the damage factor according to Equation (10), which reflects the degree of coal damage.

Since the press machine will stop working when reaching the pre-set damage conditions, it will result in the incomplete destruction of the coal specimens, which makes it impossible for the damage factor to reach 1. As a result, the damage factor can be amended as Equation (11) [29].

$$\frac{D}{D_u} = \frac{Q_d}{Q'_m} \tag{11}$$

where $D_u$ is the critical value of damage and $Q'_m$ is the charge accumulation when the damage factor reaches the corresponding value.

The critical value of damage in Equation (11) can be normalized with stress linearization, which is obtained by Equation (12).

$$D_u = 1 - \frac{\sigma_r}{\sigma_c} \qquad (12)$$

where $\sigma_c$ and $\sigma_r$ are the peak strength and residual strength of the coal specimen, respectively.

The revised damage factor $D_r$ can be expressed as Equation (13) by substituting Equation (12) into Equation (11).

$$D_r = \left(1 - \frac{\sigma_r}{\sigma_c}\right) \frac{Q_d}{Q'_m} \qquad (13)$$

According to Equation (13), the curves of stress–strain and charge accumulation strain of coal specimens under uniaxial compression conditions are as shown in Figure 7. At the beginning of stress loading, the charge accumulation curve shows a low-amplitude linear increasing trend, and the damage degree of the coal mass develops at a constant speed. During the plastic stage, the charge accumulation curve jumps in a stepwise manner, especially when the stress of the coal specimen is near the peak value. The rapid charge increase indicates that the damage degree of coal mass accelerates. Within the failure stage, the level of charge accumulation jumps the most, showing the highest degree of coal mass damage. As the coal specimen gradually approaches the residual strength point, the increased rate of charge accumulation slows down until it reaches the top, indicating that the main damage has happened.

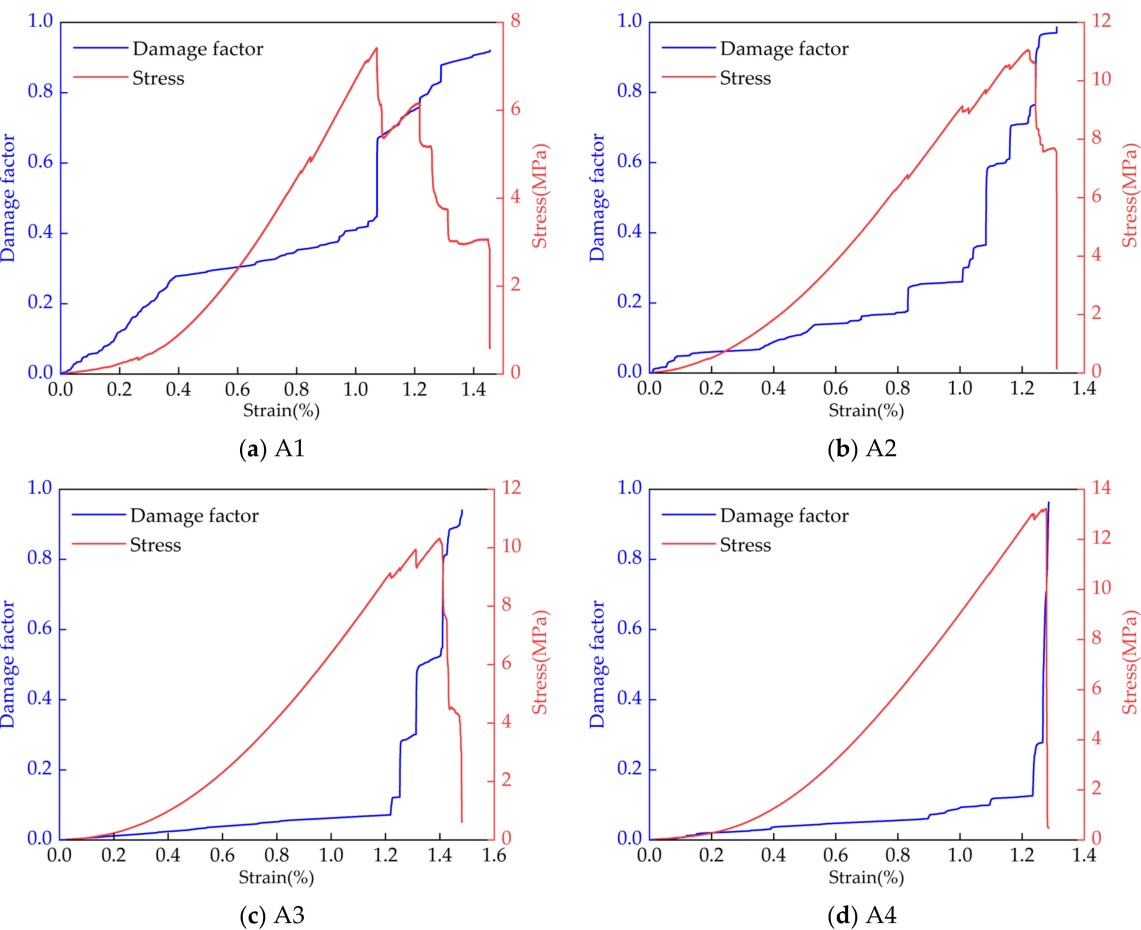

**Figure 7.** Damage factor–stress–strain curve of coal samples under uniaxial compression at different loading speeds.

### 6.2. Relationship between Coal Damage and Signal Multifractal Characterization Parameters

The induced charge signal is segmented into multiple subsets based on the degree of damage and destruction, followed by statistical analysis to examine the variations in the multifractal parameters $D(q)_{max}$, $\Delta\alpha$, and $\Delta f$ of each subset, thereby exploring their damage response. Figure 8 illustrates the multifractal characteristic curve of the induced charge for each specimen under different degrees of damage. The horizontal axis denotes the damage factor of the coal body, while the three vertical axes represent the generalized dimension $D(q)$, fractal spectrum width $\Delta\alpha$, and fractal spectrum height difference $\Delta f$ from top to bottom.

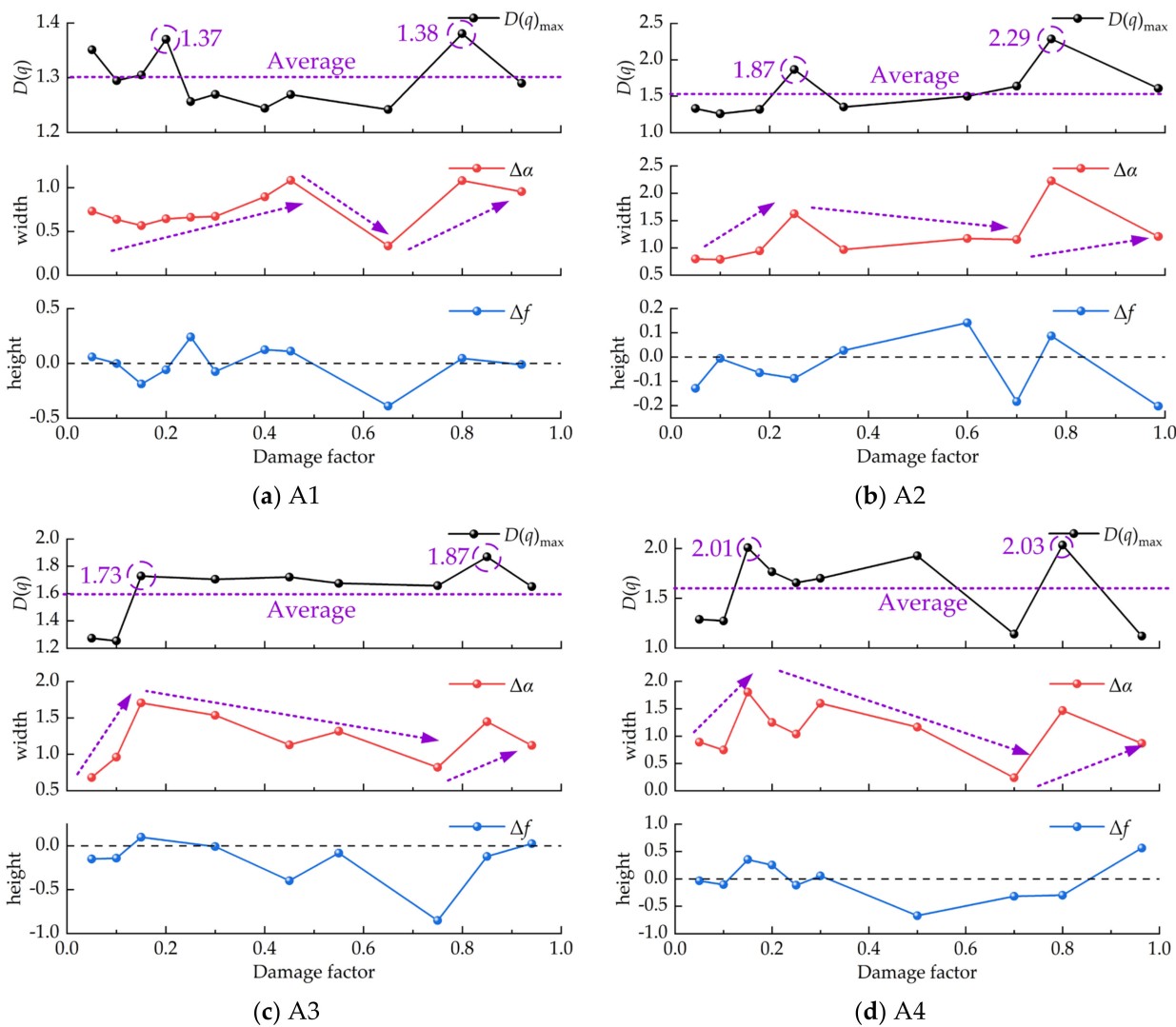

**Figure 8.** Multifractal characteristic curve of induced charge signals of coal samples under different damage degrees.

In terms of the generalized dimension, the $D(q)_{max}$ reflects the non-uniformity of the amplified subset of low-value charge signals. As the internal damage of the coal continues developing, it becomes apparent that the maximum generalized dimension $D(q)_{max}$ of the induced charge signal waves near its average line, yet significant jumps are observed in the early and late stages of damage. These jumps indicate that the coal has experienced relatively high energy damage and fracture events. With increasing loading rates, the maximum generalized dimension $D(q)_{max}$ of the signal can reach maximum values such as 2.03 and 2.29 (as shown in Figure 8b,d), which are 21% and 34% higher than the average value, respectively. This suggests that the jump amplitude of the maximum generalized

dimension $D(q)_{max}$ can represent the degree of damage and fracture of the coal. The larger the jump amplitude, the more extensive the internal fracture events that occur.

Additionally, the fractal spectrum width $\Delta\alpha$ and the fractal spectrum height difference $\Delta f$ exhibit the distinct relationships with coal damage development:

(1) When the damage degree is low and the specimen is in the compaction stage with minimal plastic damage, the induced charge signals are primarily influenced by the piezoelectric effect. Due to variations in the loading speeds and the development of internal microcracks, induced charge signals vary in richness. Primary coal fractures exhibit significant irregularity in scale, type, and spatial distribution, resulting in notable differences in the amplitude, frequency, and continuity of induced charge signals. Consequently, $\Delta\alpha$ shows an upward trend, typically ranging between 0.5 and 1.0 initially. $\Delta f$ fluctuates below the zero line, indicating a relatively higher occurrence of induced charge signals at slightly higher values.

(2) As the stress level continues to increase and the sample enters the elastic stage, the evolution of internal damage tends to stabilize. The accumulation of elastic energy and the stable crack expansion lead to a slight increase in the induced charge amplitude. Occasionally, localized damage and failure events may occur, causing $\Delta\alpha$ to continue its upward trend, reaching approximately 1.5. Meanwhile, the fluctuation degree of $\Delta f$ expands further, indicating a change in the dominance of high-value induced charge signals with varying degrees of damage.

(3) With the stress level rising further and the specimen entering the plastic deformation stage, the damage and failure intensify. Phenomena such as friction electrification, crack propagation, and coal particle ejection result in an increase in the induced charge amplitude and response frequency. High-value charge signals often dominate during this period, leading to a downward trend in both $\Delta\alpha$ and $\Delta f$. The range of decrease in $\Delta f$ is relatively small in stages (a) and (b), but as the damage value approaches 0.6, $\Delta f$ drops below zero, indicating a significant influence of the loading rate on the damage evolution process of the coal body, resulting differences in induced charge richness.

(4) In the failure stage, as the damage degree steadily increases and gradually reaches its peak, the coal continues to produce charges of varying amplitudes with adjustments to the stress structure. $\Delta\alpha$ exhibits a sharp rise and fall trend, while the fluctuation of $\Delta f$ generally increases.

Overall, the multifractal spectral width $\Delta\alpha$ of induced charge signals for coal damage and failure exhibits a trend of increasing–decreasing–increasing. Meanwhile, $\Delta f$ mainly fluctuates around zero during the early stage of damage development. However, as severe damage and destruction occur, the fluctuation degree of $\Delta f$ increases, indirectly reflecting the intensity of the signal.

## 7. Discussion

### 7.1. Accuracy and Efficiency

The research conducted in this paper reveals that the multifractal parameters of induced charge signals undergo significant changes corresponding to the coal damage. Notably, during high-impact rupture events, there is a distinct jump observed in the maximum multifractal dimension $D(q)_{max}$, accompanied by a corresponding increase in the multifractal spectrum width $\Delta\alpha$. These findings suggest a strong correlation between coal damage severity and multifractal signal characteristics.

Furthermore, a review of the existing literature, including studies on microseismic, AE, EMR, and other signals was consistent with our research findings [19,21,25]. These studies also document similar shifts in multifractal characteristic parameters in response to dynamic processes, such as structural deformation and failure.

Moreover, while the applicability of laboratory-scale observations in underground large-scale induced charge signal monitoring may be limited, the inherent self-similarity of multifractals offers a promising avenue to bridge the gap. By leveraging multifractal parameters, we can effectively discern precursory signals indicative of coal–rock dynamic disasters.

In summary, our findings contribute to the growing body of evidence supporting the efficacy of multifractal analysis in elucidating the intricate dynamics of coal–rock systems. By demonstrating the consistency of multifractal parameter shifts across different signal modalities and dynamic processes, our research underscores the robustness of multifractal methods in capturing and characterizing complex behaviors in various scientific and engineering contexts.

### 7.2. Research Limitations

Since our study primarily focuses on the multifractal characteristics of induced charge signals during the coal damage and failure process, further refinement of the subsets based on the degree of damage can enhance the robustness of the article.

Furthermore, to augment our analytical approach and further advance research in the field of coal–rock dynamic disasters, integrating additional methods such as artificial intelligence (AI) and novel information technologies is essential. AI techniques, including machine learning algorithms and deep learning models, can offer valuable insights into complex datasets and facilitate predictive analysis of coal damage and failure [30]. Additionally, multi-criteria decision-making models [31] and the entropy weight method (EWM) [32] can help prioritize risk factors and optimize decision-making processes in disaster prevention and mitigation strategies.

By leveraging a diverse range of analytical methods, we can enhance the comprehensiveness and accuracy of our research findings, paving the way for more effective monitoring and management of coal–rock dynamic disasters in mining operations.

## 8. Conclusions

This study proposes a multifractal analysis method to evaluate the variations in induced charge signals, providing multifractal characterization as the indicator of coal–rock dynamic disasters. The proposed approach was tested using coal specimens in Fuxin province, China. The main conclusions are drawn as follows:

(1) The amplitude and fluctuation of induced charge signals are stronger in the late loading stage compared to the early stage. With an increase in the loading rate, more abundant signals will be generated in the late stage.

(2) The multifractal characterization parameters of induced charge signals can reflect the process of coal damage and failure. Increases in the loading rate and loading degrees improve the $D(q)_{\max}$ and $\Delta\alpha$, providing a method for early warnings of coal instability and failure.

(3) The maximum generalized dimension $D(q)_{\max}$ of the induced charge signal before and after the coal damage process exhibits a noticeable jump. A larger jump indicates a richer occurrence of coal damage and fracture events. The spectrum width $\Delta\alpha$ exhibits an overall trend of increase–decrease–increase, while $\Delta f$ fluctuates around zero in the early stages of damage development and increases significantly during severe damage and destruction events, reflecting the intensity of the signal.

**Author Contributions:** Conceptualization, J.L. and S.L.; data curation, S.L.; formal analysis, S.L.; funding acquisition, J.L.; investigation, Z.T.; methodology, J.L. and Y.P.; project administration, J.L. and Y.P.; resources, J.L.; software, J.L. and S.L.; supervision, Y.P. and Z.T.; validation, S.L.; visualization, S.L. and Y.P.; writing—original draft, S.L.; writing—review and editing, J.L. and Z.T. All authors have read and agreed to the published version of the manuscript.

**Funding:** The research was supported by the National Natural Science Foundation of China (Grant No. 52374205), the Hebei Key Laboratory of Mine Intelligent Unmanned Mining Technology (North China Institute of Science and Technology) (Grant No. iium007), and the Fundamental Research Project of the Educational Department of Liaoning Province (Grant No. JYTMS20230793).

**Institutional Review Board Statement:** Not applicable.

**Informed Consent Statement:** Not applicable.

**Data Availability Statement:** The data that support the findings of this study are available from the corresponding author upon reasonable request.

**Acknowledgments:** We would like to express our sincere gratitude to the editors and reviewers who have put considerable time and effort into their comments on this paper.

**Conflicts of Interest:** The authors declare that they have no known competing financial interests or personal relationships that could have appeared to influence the work reported in this article.

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
