# Peer review of "Promoting Sustainable Coal Mining: Investigating Multifractal Characteristics of Induced Charge Signals in Coal Damage and Failure Process"

_sustainability, doi:10.3390/su16083127_

Round 1

Reviewer 1 Report

Comments and Suggestions for Authors

The manuscript presents laboratory-scale tests for monitoring induced charge signals of loaded coals, employing multi-fractal analysis methods to obtain its multi-fractal law during the coal damage and failure process, studying the variations of multi-fractal characteristic parameters under different damage levels. This paper has certain innovations on the multi-fractal characteristics of induced charge signals and their relationship with coal damage and failure.

Overall, the paper will be accepted after minor revisions. However, the following problems still need to be corrected or explained:

1.      In the first paragraph of the Introduction, methods for monitoring and early warning of coal and rock dynamic disasters are explained, but some are not mentioned. Please supplement them.

2.      The second paragraph of the Introduction mainly explains the relevant research on induced charges in coal and rock, but lacks an internal connection with the third paragraph, which should be supplemented.

3.      There are two references in the text that displays "error! Reference source not found", please correct them.

4.      The relevant physical parameters of the experimental coal samples are not given. Please provide them.

5. The experiment was conducted using displacement loading at different rates. What was the basis for selecting the rates?

Comments on the Quality of English Language

The quality of English laguage is needed to be improved.

Reviewer 2 Report

Comments and Suggestions for Authors

Overall, the manuscript is well-organized, and the context of this study is clearly stated. It presents several notable innovations, particularly using multifractal theory to analyze the time series of induced charge, and the multifractal index and its variation trend of induced charge signals in the coal damage and failure process are proposed. The paper reaches its objective and provides reasonable results, which can enhance understanding of the early warning of coal-rock dynamic disasters.

However, there are still some issues in this manuscript that need to be improved. After the revision, it can be accepted. The following are the specific issues:

1.       Line 186, what is the scientific basis for setting 25mm and 1KHz?

2.       There are reference errors in Line 194 and Line 220, please correct them.

3.       Line 245: "This suggests that high-value induced charge signals hold an absolute advantage in the late stage, and the fluctuation degree becomes more severe with the increase of loading rate." Why can the denser lower curve than the upper curve indicate that high-value signals dominate? Please add references.

4.       Experiments are carried out using processed samples of raw coal, and the experimental results obtained are usually very discrete. How did the author solve this problem in this study?

5.       Section 6.1, the definition of damage factor is introduced without references, please add it.

Reviewer 3 Report

Comments and Suggestions for Authors

The paper was well written and the results were nicely presented. I only have some minor comments as given below:

The testing condition should be explained in more detail. Was the test stress-controlled or displacement-controlled? 

A cross citation error occurs in lines 194 and 195. Please check.

You only presented the data in double log scale but did not do curve fitting. So, the caption of Figure 4 should be modified accordingly.

How to distinguish between early-stage and late-stage? Please explain.

Reviewer 4 Report

Comments and Suggestions for Authors

The research aims to carry out the monitoring test for induced charge signals of coal under different loading rated and used the multifractal theory to analyze its statistical multifractal patterns. Also, for studying the trend changes during the loading process, multifractal characterization parameters were utilized. For describing the evolution degree of multifractal characteristics parameters of induced charge signals with coal damage and failure, combined with damage mechanics theory, a characterization relationship based on induced charge accumulation was proposed. In general, this research offers theoretical insights for the practical monitoring of coal-rock instability and failure. I read this manuscript carefully and the content is interesting. However, this manuscript needs to be carefully revised before it can be accepted. The following comments will be useful for enhancing the quality of this paper.

v Abstract:

·         Please revise the abstract from Lines 11-14. This section is not clear

·         Please double-check the abstract in terms of grammatical error. Also, please connect the sentences to each other.

·         It would be better to replace the symbols in the abstract with their original type (Lines 25, 28)

·         Rewrite the keywords: write the keywords according to Alphabet (This will help others to search your paper easily). Also, please pay attention; the first word of keywords should be written in Caps Lock

·         There are too many words and each keyword should a maximum of two words (Line 34)

·         It would be better to add the name of study area as the location of the test system in the abstract, the last paragraph of the introduction and also in the conclusion.

v  Introduction:

·         Please summarize the introduction (from Lines 52-96); please select some key research in this field study

·         I see so few papers from the last years. Please cite the recent papers (2023-2024) as much as possible. It helps to readers get familiar with recent literature on this subject.

·         Please double-check the introduction in terms of Grammatical and typing errors (Lines 98 and 99; this sentences are not clear)

·        Please diversify your citation; too many citations to your country. Please mention some related research from other parts of the world.

·         Please mention the novelty of your research in the last paragraph of the introduction (Lines 98-109).

v Section 3:

·         It would be better to mention the loading rates in the Line 173

·         How many sample you gathered from the study area? Please mention the details

·         How did you gather the data? It should be clear for readers and also help to the repeatability of your research.

·         It would be better to insert a table and mention the number of samples, the ranges of loading stress and etc.

·         What is the DC in Line 187? Please make sure to write the full word of abbreviations for the first in the whole manuscript.

·         What is the meaning of the last sentence of Line 194? Please double-check this section

·         In the Line 193; the name or related information of the 4 groups should be clear.

v Section 4

·         Please double-check the Line 220 what is the "Error! Reference source not found"?

v Section 5

·         Please use a same format for mentioning the Figures and Tables in the manuscript (Line 235)

·         Please write the related units for the numbers in the Lines 235 and 236 and also Line 353 in section 6

v Conclusion:

·        The conclusion is too long. Please summarize the conclusion in short sentences. It should include home-taking messages.

·         Please write the details loading test and different rates in your manuscript clearly so that everybody can find the information at a glance. In general, please write your methodology briefly in the conclusion and focus on your result

·         Please write the conclusion as a single paragraph

·         It is necessary to highlight the novelty of your research in the conclusion

·         Please add a section before the conclusion and write about the limitations and also insert a section before, within ,or after the conclusion for recommendations for future studies

v  Specific comments:

·         It would be better to edit the title of section 5 and 6; this help readers to find your findings easily

·         Please make a comparison between your findings and the result of other research in a separate section with the title of discussion.

v  Limitation should be discussed through comparison with other methods, e.g., risk assessment [DOI: 10.1007/s44268-023-00020-4], [DOI: 10.1007/s44268-023-00011-5], [doi: 10.1007/s44268-023-00002-6.].

Comments on the Quality of English Language

significant English edit should be conducted.

Round 2

Reviewer 4 Report

Comments and Suggestions for Authors

no comments.